# Public perception of COVID-19 management and response in Nigeria: a cross-sectional survey

Obinna Oleribe,[1,2] Oliver Ezechi,[3] Princess Osita-Oleribe,[4] Olatayo Olawepo,[5] Adesola Z Musa,[3] Anddy Omoluabi,[6] Michael Fertleman ,[7] Babatunde L Salako,[8] Simon D Taylor-Robinson[9]

For numbered affiliations see end of article.

**Correspondence to**
Dr Michael Fertleman;
m.fertleman@imperial.ac.uk

## ABSTRACT

**Objectives** A study designed to assess the public perception of the response of government and its institutions to the COVID-19 pandemic in Nigeria.

**Setting** Self-selecting participants throughout Nigeria completed a self-administered questionnaire through an online cross-sectional survey.

**Participants** 495.

**Results** The majority of respondents were married (76.6%), were males (61.8%), had tertiary level education (91.0%), were public servants (36.8%), Christians (82.6%), and resident either in the Federal Capital Territory (Abuja) (49.1%) or in the South-East Region of Nigeria (36.6%). Over 95% of the respondents had heard of COVID-19 (98.8%) and knew it is a viral disease (95.4%). The government and its institutions response to the pandemic were rated as poor, with the largest rating as poor for Federal President's Office (57.5%). Communication (50.0%) and prevention messages (43.7%) received the highest perception good rating. Female respondents and those less than 40 years generally rated the governmental responses as poor.

**Conclusions/recommendations** It is recommended that as a public–private partnership approached was efficiently used to more effectively disseminate public health communication and prevention messages, the Nigerian Government should expand this collaboration to improve the quality of services provided in other areas of COVID-19 outbreak management.

## INTRODUCTION

There have been fundamental changes to society with the emergence of the SARS-CoV-2 global pandemic.[1–4] As COVID-19 has no specific antiviral drug treatment or vaccine to date, universal safety precautions and mitigating strategies are the only common way to deal with this global health emergency.[4]

As COVID-19 outbreak posed significant challenges for the public health, research and medical communities, governmental policies and processes were introduced to curtail the spread of the virus across the nations of the world.[5] This included instituting temporary restrictions on travel with dramatic reduction in the number of travellers, self distancing,

## Strengths and limitations of this study

► First African study of public perception of government response to the COVID-19 pandemic.
► Study was cross-sectional and questionnaire was self-administered, thereby reducing researcher-induced bias but potentially increasing selection bias.
► Study provides perspectives from different classes of Nigerians on the COVID-19 pandemic.
► While providing evidence for informed decisions by national leaders and programme managers, the study period was short and therefore limited.
► Only individuals with social media accounts or internet access could participate in the study.

self isolation, quarantine of international travellers, regular hand washing and the use of face masks.[5–9] These policies have their economic implications to both developed and developing nations, such as Nigeria.[10]

In Nigeria, most of these global policies were adopted and implemented without a review of their effectiveness and implications to the sociocultural climate of the nation. For instance, physical distancing measures may save many lives in high-income countries, but are less effective in poor countries with younger populations who are less susceptible to COVID-19.[11] Such measures are associated with healthcare systemic, economic and sociocultural challenges.[10]

People living in poverty are less willing to make economic sacrifices, not because they place greater value on their livelihood concerns than contracting COVID-19,[10] but because they lack the resources and social protections to isolate themselves and sacrifice economic opportunities.[10] The country formulated new policies, developed new guidelines and standard operating procedures, and established several committees including the National Emergency Organizing Committee and Presidential Task

Force (PTF)[12 12 13] to respond to the pandemic. The governmental efforts at all levels were complemented by donations and contributions from the organised private sector, health professional associations, non-governmental organisations (NGOs) and individual volunteers.[14–16]

However, the pandemic in Nigeria continues unabated despite the various efforts of individuals, government and organised private sector with 46 803 tested, 8344 confirmed cases and 249 deaths in 3 months (27 February to 27 May 2020) of the pandemic in Nigeria,[12] prompting plans to introduce addition measures.

The Nigerian President established the PTF on COVID-19 to manage the outbreak of disease across the country. The PTF was mandated to work with the Federal Ministry of Health (FMOH) and the Nigerian Center for Disease Control (NCDC) in the implementation of the various initiatives and strategies on COVID-19 containment. While FMOH and NCDC were responsible for the implementation of strategies to track the epidemic, to develop management guidelines and to engage and train field workers, the PTF was responsible for local and international relationships, mobilisation and management of resources, together with weekly briefings of the President and the Federal Nigerian Government. State and local governments also established similar COVID-19 structures that were all answerable to the local State governments of the Nigerian Federation.

The public perception, opinion and attitude to the various social control interventions introduced in the wake of the pandemic have not previously been evaluated in Africa, although some work has been produced around public perception in Europe[17] and the USA.[18] Information obtained may be useful in planning, modifying and implementing a coordinated response to current and future epidemics in Nigeria. Since it is established that public perception and attitude are important for programme success, this study was designed to assess, document and evaluate public knowledge and perception of the measures introduced to combat the COVID-19 pandemic in Nigeria.

## METHOD
### Timeline
This study was conducted between 15 May 2020 and 21 May 2020 inclusive, through a nationwide, anonymous online cross-sectional survey.

### Participants
Participants were obtained using a self-selecting sampling technique. The participants were informed of the anonymity of their participation and that information collected would be kept confidential. Only participants 18 years and above were invited to participate in the survey.

### Sample size
We calculated an approximate minimum sample size of 452 at 5% precision, 95% CI and 50% response distribution (online survey) (http://www.raosoft.com/samplesize.html).

### Patient and public involvement
Between 10 and 12 May 2020, the tool was developed. Face validity was established by trialling the survey on different members of staff from the first author's institution. Between 13 and 15 May 2020, the tool was reviewed with seven members of the public at the lead author's institution, before finalising with their input. A small test pilot was carried out and questions were then refined by the first authors. Further validation was limited, owing to the timeframe of the study.

No patients were involved in the design of the study.

### Survey
Google Form (https://docs.google.com/forms/d/1P36 vWQ0Ihf8HMTLHWIF83aZy-hFKVla7FSSm-WZudMY/edit) was used to develop and distribute the questionnaire. Between 10 and 12 May 2020, the tool was developed. Between 13 and 15 May 2020 the tool was reviewed with seven members of the public, before finalising with their input. The study was designed as an online survey to cover the entire country during the 7-day period. The questionnaire was in English and was pretested for comprehensibility, acceptability and accuracy before the study was commenced. Invitations to participate in the study were through private messages, social groups and on several social networking sites and platform (emails, WhatsApp and Facebook). By clicking on a link, the participant was directed to the survey's entry page (https://docs.google.com/forms/d/e/1FAIpQLSf4iuWGfO4pIvSVi24Sn0 DhZSO9q_wnpYhB2J3boAsQ7W4Uww/viewform?vc=0&c=0&w=1&usp=mail_form_link), which contained information on the objectives of the survey, terms of participation and data privacy. Participants were also informed about the possible risks and benefits of the survey. They were asked to complete the survey in one session and that it would take between 10 and 15 min. Participants were able to access the survey and complete it on a computer or a mobile device.

The survey consisted of 30 semistructured questions. The questionnaire collected information on sociodemographic characteristics, knowledge of COVID-19 and individual perspective on management and response to the pandemic. The questions were semistructured as some questions allowed participants to provide detailed written answers to obtain a more profound sense of their perspective. The data collected for the survey and used in this study were stored electronically and were password protected.

### Measures
#### Sociodemographic characteristic
Participants responded to closed-ended demographic questions including their gender, age, relationship status, city and state of residence, work status and educational status.

#### Response to COVID-19
Participant perception of the measures introduced by the Federal Government and governmental institutions

to COVID-19 pandemic were assessed in the survey. The respondent perception of scores ranging from the lowest score of 1 (not satisfied at all) to the highest score of 10 (very satisfied) was further classified as poor (score of 1–4, average (score of 5–7) and good (scores of 8–10).

## Data analysis
Data collected on Google Analytic Tool were exported to and analysed with the Statistical Package for Social Sciences for Windows V.23.0 (SPSS, Chicago, Illinois). A p value of <0.05 was considered statistically significant.

## RESULTS
A total of 495 respondents returned completed questionnaires during the study period. While 482 (97.4%) respondents were resident in Nigeria, 13 (2.6%) were temporarily resident outside the country owing to border closures. The age of the respondents ranged from 18 to 59 years, with an average age of 42.1 years (SD=9.7). The majority of respondents were married (76.6%), males (61.8%), had tertiary level education (91.0%), were public servants (36.8%), Christians (82.6%), and resident either in the Federal Capital Territory (49.1%) or the South-East region (36.6%) (table 1). Over 95% of the respondents had heard of COVID-19 (n=489, 98.8%), and knew that it is a viral disease (n=472; 95.4%).

Table 2 shows the respondent perception of the response of the Federal Government and the governmental institutions to the COVID-19 outbreak in Nigeria. The government measures were rated as poor, with the Office of the Presidency having the highest poor rating score by 57.5% of the respondents. The FMOH, the Presidential COVID-19 Task Force and the Nigerian Centre for Diseases Control responses to the outbreak were rated as poor by 46.1%, 45.8% and 36.0%, respectively. The respondents perception of specific governmental and institutional responses to COVID-19 are detailed in table 3. Among the specific responses by Government and its institutions to the pandemic, public health communication about the pandemic and general prevention messages received the highest good rating by 50.0% and 43.7% of the respondents, respectively. The availability of social and economic support from the Federal Government (n=406; 82.9%) had the highest poor rating score, followed by governmental management of the resources provided for the outbreak (n=368; 75.4%) and readiness of the government to manage the pandemic (65.5%). Social and economic support from religious bodies and family and friends were rated as poor by 52.1% and 33.1% of the respondents, respectively.

The respondent perception of governmental and institutional response by respondent age group and sex is shown in table 4. The proportion of respondents less than 40 years that rated the governmental and institutional measures to combat COVID-19 as poor

**Table 1** Sociodemographic characteristics of respondents in the study

| Characteristics | Respondents, n, (%) |
|---|---|
| Country of residence | |
| Nigeria | 482 (97.4) |
| Others | 13 (2.6) |
| Geopolitical zone of residence | |
| North-Central | 243 (49.1) |
| North-East | 29 (5.9) |
| North-West | 30 (6.1) |
| South-East | 68 (13.7) |
| South-South | 39 (7.9) |
| South-West | 86 (17.4) |
| Marital status | |
| Single | 100 (20.2) |
| Married | 379 (76.6) |
| Divorced/separated | 9 (1.8) |
| Widowed | 7 (1.4) |
| Age (years) | |
| Less than 20 | 1 (0.2) |
| 20–29 | 61 (12.3) |
| 30–39 | 142 (28.7) |
| 40–49 | 159 (32.1) |
| 50–59 | 93 (18.8) |
| 60 years and above | 39 (7.9) |
| Sex | |
| Female | 189 (38.2) |
| Male | 306 (61.8) |
| Religion | |
| African traditional religion | 2 (0.4) |
| Christianity | 409 (82.6) |
| Islam | 79 (16.0) |
| Others | 3 (0.6) |
| Education qualification | |
| Non formal | 18 (3.6) |
| Primary | 2 (0.4) |
| Secondary | 20 (4.0) |
| Tertiary | 455 (91.9) |
| Work status | |
| Working | 338 (87.5) |
| Not working | 38 (7.7) |
| Student | 24 (4.8) |

was greater than those 40 years and above across all aspects of the survey. However, it is only in the perception of the Office of the Presidency that the difference was statistically significantly higher (p=0.03). The proportion of female respondents who rated the

**Table 2** Respondent perception of the governmental and governmental institutional response to the COVID-19 pandemic in Nigeria

| Government /institution | Respondents COVID-19 response perception rating | | |
| --- | --- | --- | --- |
| | Poor, n, (%) | Average, n, (%) | Good, n, (%) |
| The Presidency (N=487) | 280 (57.5) | 130 (26.7) | 77 (15.8) |
| Presidential Task Force on COVID (N=491) | 225 (45.8) | 168 (34.2) | 98 (20.0) |
| Federal Ministry of Health (N=490) | 226 (46.1) | 164 (33.5) | 490 (20.4) |
| Nigerian Center for Disease Control (N=492) | 177 (36.0) | 157 (31.9) | 158 (32.1) |

governmental measures as poor were higher compared with their male counterpart across all aspects of the survey. The observed difference was only statistically significantly higher again in the perception of the response of the Office of the Presidency (p=0.05).

A spatial dimension of the responses was also analysed by aggregating the ratings of the respondents by their state of residence with the results presented in figure 1. Responses were from 33 states including the Federal Capital Territory (n=33). Respondents from 10 states (30%) rated the response of government as good, respondents from 12 states (36%) rated the governmental response as average, while the respondents from the remaining 11 states (33%) rated the responses as poor. Analyses of the rating by states, indicated that Lagos states, even though it has the highest number of laboratory confirmed COVID-19 cases in the country 3756 (45%), rated the government response as good. This contrasts with Kano state, which comes second with 923 (11%) confirmed cases, but rated the governmental responses as poor. The FCT, which comes third in terms of confirmed cases 519 (6%), rated the government response as average.

To determine the statistical significance of the relationship between the responses and the number of confirmed COVID-19 cases, the Pearson's correlation was applied. A Pearson's coefficient of 0.11 and a p value of 0.51 were obtained, indicating a weak and insignificant relationship between the responses and number of confirmed COVID-19 cases.

## DISCUSSION

The success and sustainability of public health programmes are highly dependent on the positive perception and acceptance by the general public. Risk communication associated with a particular hazard involves the exchange of information and advice between experts and the public, as it becomes available. The ultimate aim of risk communication is to permit people at risk to take informed decisions to protect themselves.[19] The continued increase in new cases of COVID-19, 3 months after the first case, was reported in Nigeria on 27 February 2020 has made Nigerians wonder about the competency of the government response to the pandemic. As a result of this failure to cap the outbreak, significant numbers of

**Table 3** Respondents perception of the governmental and governmental institutional-specific response to COVID-19 outbreak in Nigeria

| Specific response | Respondents COVID-19 response perception rating | | |
| --- | --- | --- | --- |
| | Poor, n, (%) | Average, n, (%) | Good, n, (%) |
| Enforcement of stay at home, physical distancing, face mask and handwashing policies (N=492) | 244 (49.6) | 141 (28.7) | 107 (21.7) |
| Management of isolation centres (N=482) | 265 (55.0) | 148 (30.7) | 69 (14.3) |
| Management of treatment and diagnostic centres (N=477) | 254 (53.2) | 150 (31.4) | 73 (15.3) |
| Communication and information about the epidemic (N=490) | 129 (26.3) | 116 (23.7) | 245 (50.0) |
| Prevention messages from government (N=490) | 146 (29.8) | 130 (26.5) | 214 (43.7) |
| Use of personal protectiveequipment by healthcare workers (N=484) | 232 (47.9) | 134 (27.7) | 118 (24.4) |
| Availability of social and economic support from government (N=490) | 406 (82.9) | 53 (10.8) | 31 (6.3) |
| Availability of social and economic support from religious bodies (N=486) | 253 (52.1) | 129 (26.5) | 104 (21.4) |
| Availability of social and economic support from family and friends (N=487) | 161 (33.1) | 158 (32.4) | 168 (34.5) |
| The readiness of the government to manage the epidemic (N=490) | 321 (65.5) | 102 (20.8) | 67 (13.7) |
| Government management of the resources provided for the outbreak (N=488) | 368 (75.4) | 82 (16.8) | 38 (7.8) |

**Table 4** The respondents perception of government and institution response to COVID 19 pandemic in Nigeria by sex and age-group

| | Respondents COVID-19 response perception rating | | | | | | |
|---|---|---|---|---|---|---|---|
| | Poor | Average | Good | Poor | Average | Good | P value |
| Government/institution | Male | | | Female | | | |
| The Presidency (N=487) | 167 (59.6) | 80 (61.5) | 56 (72.7) | 113 (40.4) | 50 (38.5) | 21 (27.3) | 0.05 |
| Presidential Task Force on COVID-19 (N=491) | 130 (57.8) | 112 (66.7) | 62 (63.3) | 95 (42.2) | 56 (33.3) | 36 (36.7) | 0.19 |
| Federal Ministry of Health (FMOH) (N=490) | 128 (56.6) | 115 (70.1) | 60 (60.0) | 98 (43.4) | 49 (29.9) | 40 (40.0) | 0.23 |
| Nigerian Center for Disease Control (NCDC) (N=492) | 104 (58.8) | 102 (65.0) | 99 (62.7) | 73 (41.2) | 55 (35.0) | 59 (37.3) | 0.44 |
| | Less than 40 years | | | 40 years and above | | | |
| The Presidency (N=487) | 126 (45.0) | 51 (39.2) | 25 (32.5) | 154 (55.0) | 79 (60.8) | 56 (72.7) | 0.03 |
| Presidential Task Force on COVID-19 (N=491) | 98 (43.6) | 70 (41.7) | 35 (35.7) | 127 (56.4) | 98 (58.3) | 63 (64.3) | 0.21 |
| FMOH (N=490) | 92 (40.7) | 73 (44.5) | 38 (38) | 134 (59.3) | 61 (55.5) | 62 (62.0) | 0.83 |
| NCDC (N=492) | 68 (78.4) | 70 (44.6) | 66 (41.8) | 109 (61.6) | 87 (55.4) | 92 (58.2) | 0.51 |

the public are disillusioned and are not keeping to the government-issued guidelines and recommendations, with some believing that the COVID-19 pandemic is a hoax. This study was conducted to evaluate the perceptions of Nigerians to the measures taken by the Federal Government and its governmental institutions in response to the COVID-19 outbreak in the country.

In this study, the majority of the respondents were resident in Nigeria, had tertiary level education, were males having heard of COVID-19 and knew its viral aetiology (95.4%). The study shows that the respondents were educated and had some level of knowledge of COVID-19, and thus could form a meaningful opinion on the governmental response. The

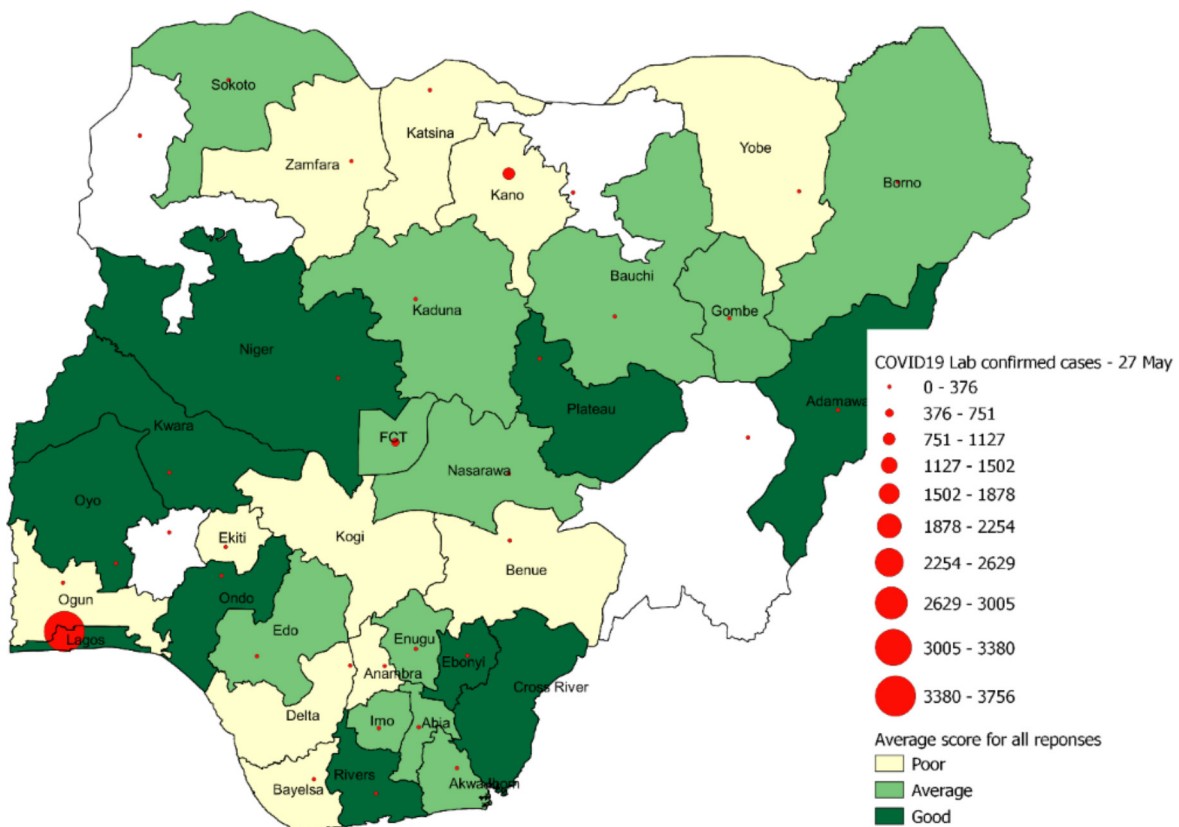

**Figure 1** Spatial distribution of respondents to COVID-19 response and management in Nigeria.

respondents' residency in Nigeria highlights that their opinions are well informed and could also impact on how the government guidelines and recommendations might be observed by the general public. The high educational attainment of the respondents may be a reflection of their awareness of importance and the understanding of the relevance of research. The male preponderance in this study may be a reflection of previously observation phenomena that gender plays a role in participation in online surveys.[20]

The study shows that the respondents were unhappy with the manner of responses provided by the Federal Government and its agencies including the Office of the Presidency, the PTF on COVID-19, the FMOH and the NCDC. Apart from NCDC, over 70% of all respondents scored the Office of the Presidency, PTF and FMOH poor or average with 57% scoring the Office of the Presidency as poor in its performance. This may be an indication of either a delayed or an inadequate response to the crisis, which may have contributed to the increasing number of new cases of COVID-19 and deaths in Nigeria 3 months after the first case was reported.

Similarly, the majority of the respondents rated specific individual measures by the government and its institutions as poor. Over 50% of the respondents scored the management of isolation, and treatment/diagnostic centres as poor. This is in support of the numerous social media videos and comments against these centres across Nigeria. Communication and delivery of prevention messages accrued better scores. This specific area of public health messaging was provided through a public–private partnership between the government agencies and global system for mobile technology operators in the country.

Unfortunately, the enforcement of the implementation of the guidelines and policies are still poor. Mainly because of the negative perception of governmental measures taken, individuals have not taken the pandemic seriously. The situation is compounded by the presentation of COVID-19 in Nigeria where a significant number of cases have been asymptomatic and the case fatality rate has been low[21] making people believe that the disease is not a serious illness.[22]

Most social and economic support came from family and friends, followed by the religious bodies. The majority (over 80%) believed that government social welfare provisions were poor. One cannot, however, identify the root cause of this perception as funds were made available for supportive measures for sectors of the population in Nigeria. Whether the funds were not adequately released, released but diverted or poorly distributed remains to be seen. This calls for new in-depth studies to identify the root cause of this unsatisfactory aspect of governmental performance during this crisis.

Over 65% of the participants believed that the government was not prepared for the outbreak and 75% that the government poorly managed the resources provided for the outbreak. It is not, therefore, surprising to note that the government was asking returnees to pay for their quarantine or be allowed to suffer the consequences— which may include denial of access to the country.[23]

Social support, enforcement of isolation policies and provision of testing services did not go well, according to the participants. These are project management issues that the engagement of a professional project manager to operationalise the entire process would have eliminated. It is, therefore, important that skilled project managers are engaged in subsequent epidemics to mitigate poor project management, better finance management, improve quality of services and to ensure timely delivery of results and benefits.

Given a large proportion of the population under the age of 40 years is active on social media, it seems prudent to recommend that governmental policy for future epidemics should use this medium for communication to a growing sector of Nigerian society.

The findings of this online cross-sectional study have to be interpreted with some limitations in mind. The use of an online survey for this study may have resulted in a selection bias for respondents who are educated, internet savvy and have a social media account. It may have limited some category of respondents from participating. Respondents without formal education, without internet connection or who live in rural areas may have been disenfranchised. However, this will not invalidate the findings presented here, more especially since only those who are knowledgeable and know the specific governmental measures taken, can meaningfully assess what is being done. Finally, this is a cross-sectional study, and does not allow any inferences to be made regarding causality and temporality with respect to perception and the sociodemographic factors.

To our knowledge, this is among the first studies to examine the perception of governmental response by the public in West Africa. Information obtained here may be useful to guide future decisions especially when considering the impact of public perception on uptake of governmental guidelines and policies. Also, for the first time in the country, we have gained information on public perception of governmental responses during a global pandemic. Finally, this study used nationally representative data with appropriate sample size, which strengthens the external validity and generalisability of the study.

## CONCLUSIONS

The study shows a generally poor perception of governmental response to the COVID-19 pandemic by the public. All the specific interventions against COVID-19 were rated as poor except for public health communication and prevention messages. It is recommended that the public–private partnership approached used in this particular area should be adopted in the implementation all areas of COVID-19 outbreak policy to streamline perceived and actual inefficiencies.

**Author affiliations**
[1]International Office, Royal College of Physicians, London, UK
[2]Chief Executive Office, Excellence and Friends Management Care Centre (EFMC), Abuja, Nigeria
[3]Director of Research, Nigerian Institute of Medical Research, Lagos, Nigeria
[4]Center for Family Health Initiative, Kubwa, Nigeria
[5]Department of Epidemiology and Public Health, University of Maryland, Baltimore, United States
[6]Onyozar Consult, Abjua, Nigeria
[7]Cutrale Preoperative & Ageing Group, Imperial College London, London, UK
[8]Provost College of Medicine, University of Ibadan, Ibadan, Nigeria
[9]Department of Surgery and Caner, Imperial College London, London, UK

**Acknowledgements** We are grateful to the members of the Nigerian public who helped with the survey design. We acknowledge the National Emergency Outbreak Committee on COVID-19, the Presidential Task Force and the Nigerian Center for Disease Control. We celebrate all the frontline healthcare workers who put their lives at risk to control and contain this pandemic. We thank the Excellence and Friends Management Care Centre and HS3 Team for their excellent work during this period. MF and ST-R are grateful to the National Institute for Health Research (NIHR) Biomedical Facility at Imperial College London for infrastructural support.

**Contributors** OOle developed the concept, initial tool and first draft manuscript. OE and PO-O reviewed the final tool, made inputs to the original draft manuscript and supported data analysis. OOle, OO, AZM and AO did the data analysis. AO did the spatial analysis. MF, BLS and ST-R reviewed, edited and finalised the full manuscript. ST-R provided quality assurance to the project.

**Funding** This study was funded by Wellcome Trust (ISSF grant).

**Map disclaimer** The depiction of boundaries on this map does not imply the expression of any opinion whatsoever on the part of *BMJ* (or any member of its group) concerning the legal status of any country, territory, jurisdiction or area or of its authorities. This map is provided without any warranty of any kind, either express or implied.

**Competing interests** None declared.

**Patient and public involvement** Patients and/or the public were involved in the design, or conduct, or reporting, or dissemination plans of this research. Refer to the Methods section for further details.

**Patient consent for publication** Not required.

**Provenance and peer review** Not commissioned; externally peer reviewed.

**Data availability statement** No data are available. Data is available in the text with no further data available.

**ORCID iD**
Michael Fertleman http://orcid.org/0000-0003-4023-1156

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
