## [Reviewer comments · BMJ Open]

ARTICLE DETAILS

TITLE (PROVISIONAL)	Public Perception of COVID-19 Management and Response in Nigeria: A Cross-Sectional Survey
AUTHORS	Oleribe, Obinna; Ezechi, Oliver; Osita-Oleribe, Princess; Olawepo, Olatayo; Musa, Adesola; Omolabi, Anddy; Fertleman, Michael; Salako, Babatunde; Taylor-Robinson, Simon

VERSION 1 – REVIEW

REVIEWER	Elissa Abrams University of Manitoba, Canada
REVIEW RETURNED	06-Aug-2020

GENERAL COMMENTS	Major comments Abstract - - Conclusion - The conclusion is taking some leaps of faith in its message - the conclusion of this study was that risk communication was poor and that steps need to be taken to mitigate that. I am not sure this study can conclude which steps will effectively improve communication messages - this would need to be studied in an implementation science way before this conclusion could be made. Would recommend rewording conclusion Article summary - In strengths and weaknesses while the cross sectional nature reduced researcher bias it increased selection bias; this should be included Introduction - Lines 15-24 - if the goal of this study is the federal policy response I would recommend not including an entire section on medical response in the introduction. It confuses the message - Line 40 - to my knowledge it has not been determined that physical distancing measures are less effective in low-middle income countries; if this is indeed the case reference the statement - Would recommend spending some time in the introduction reviewing the political structure in Nigeria including the formal 'chain of command' to the pandemic. In the results section, when there are discussions of the federal ministry, the Nigerian CDC, the office of the presidency, the COVID19 task force, etc - it is hard to know who had which duties/tasks during the pandemic management Methods - The development and validation of the questionnaire requires some additional information. It notes the questionnaire was 'pretested' prior to commencement - how was it pretested? How
--

	were the questions selected? How were they validated? Discussion  - Line 6 - There is an entire field of risk communication - would consider incorporating some of what is known about effective risk communication, and referencing it - Lines 16-19 - would highly recommend removing the comparator to Ebola - they are very different infectious diseases - COVID19 causes much more community spread as it has a lower mortality than ebola - in my opinion an epidemic/pandemic response to COVID19 should be very different than Ebola and as this statement is conjecture would highly recommend removing - Lines 29-36 p13 - here, and throughout the discussion section, would be very careful about making statements that are unreferenced and not supported (yet perhaps) by data. It is not known that people have not taken the pandemic seriously because of governmental response. While this is certainly a possible hypothesis would state it as such and not as a foregone conclusion - unless there is evidence to support it (in which case it should be referenced) - Throughout the discussion - statements are made that can not be concluded from this manuscript - would consider significantly restructuring the discussion to focus on what this study shows, how it contributes to the literature, what was known prior to this study, and what is required next Minor comments Abstract -  - Objectives - would change 'epidemic' to 'pandemic' - Results - would reword 'highest level of poor rating' - it is confusing Introduction  - Reword the first sentence; it is too long; might consider something such as 'there have been fundamental changes to society with the emergence of the SARS-CoV-2 global pandemic...' - Line 46 - reword 'poor people' to 'people living in poverty' - Consider changing social distancing to physical distancing as recommended by the WHO throughout the document - Change epidemic to pandemic throughout the document
--	--

REVIEWER	Marco Giani King's college London
REVIEW RETURNED	08-Aug-2020

GENERAL COMMENTS	I think that the paper is interesting and well executed. Increased focus on emerging economies and the public responses to the crisis management in these areas are key. Moreover, the somewhat important limitations of the study are appropriately acknowledged. I have important comments. Firstly, since this paper is one of public opinion, it should engage with public opinion literature on COVID-19 responses including most notably already published papers and most notably Bol et al (2020) "The effect of COVID-19 lockdowns on political support: Some good news for democracy?" focusing on Europe and Gollust et al (2020) "The Emergence of COVID-19 in the US: A Public Health and Political Communication Crisis" focusing on the US. By doing so, the seemingly accurate claim that the paper is one of the first one analyzing public opinion in African countries will be contextualized.
--

	Secondly, the discussion of the results at pp 9-10 is excessively descriptive and limited to the paraphrasing of numbers. If the authors have no interpretation to suggest for why e.g. women or people in urban areas display different level of consensus, then in my way they can omit reporting these differences.
--	--

VERSION 1 – AUTHOR RESPONSE

(b) REVIEWER 1

1. Abstract

(i) Conclusion.

This has been altered according to the suggestion made

(ii) Article summary

Increase in selection bias is now listed

2. Introduction

(i) Medical response

Removed according to the suggestion made

(ii) Physical distancing measures

Reference now given

(iii) Political structure of relevant Nigerian agencies

Additional paragraph included

3. Methods

Further information on the design of the questionnaire is now given

4. Discussion

(i) Risk communication

Now included and referenced

(ii) Ebola

Comparisons removed

(iii) Link of society actions to government response

Language has been toned down / words removed as appropriate

Being a discussion we felt that some of the theme should be kept as this conclusion was based on the data from this study

(iv) Unsubstantiated claims

As per (iii) above we feel that this has been sufficiently addressed without removing some of the conclusions from our study that we still consider are appropriate.

5. Minor comments

(i) Abstract

Objectives. Epidemic changed to pandemic (and throughout document)

Results. "Highest level of poor rating" now changed

(ii) Introduction

Reword last sentence - changed as per suggestion

Reword "poor people" - changed as per suggestion

Change social distancing to physical distancing - changed as per suggestion

Change epidemic to pandemic - changed as per suggestion

(c) REVIEWER 2

1. Include public opinion Covid references from Europe and USA - added as per suggestion

2. Results excessively descriptive - we feel that the text flows sufficiently well and links to the discussion such that changes were not made

VERSION 2 – REVIEW

REVIEWER	Elissa Abrams University of Manitoba, Canada
REVIEW RETURNED	31-Aug-2020

GENERAL COMMENTS	All recommendations have been addressed
---

REVIEWER	Elissa Abrams University of Manitoba, Canada
REVIEW RETURNED	31-Aug-2020

GENERAL COMMENTS	All recommendations have been addressed.
--